# Aberrant HLA-DR expression in the conjunctival epithelium after autologous serum treatment in patients with graft-versus-host disease or Sjögren's syndrome

Katerina Jirsova[1]*, Petra Seidler Stangova[2], Michalis Palos[2], Gabriela Mahelkova[3], Sarka Kalasova[1], Ivana Rybickova[1], Tor Paaske Utheim[4,5], Viera Vesela[1]

1 Laboratory of the Biology and Pathology of the Eye, Institute of Biology and Medical Genetics, First Faculty of Medicine, Charles University and General University Hospital in Prague, Prague, Czech Republic, 2 Department of Ophthalmology, First Faculty of Medicine, Charles University and General University Hospital in Prague, Prague, Czech Republic, 3 Department of Ophthalmology, Second Faculty of Medicine, Charles University in Prague and Motol University Hospital, and Department of Physiology, Second Faculty of Medicine, Charles University in Prague, Prague, Czech Republic, 4 Department of Medical Biochemistry, Oslo University Hospital, Oslo, Norway, 5 Department of Plastic and Reconstructive Surgery, Oslo University Hospital, Oslo, Norway

* katerina.jirsova@lf1.cuni.cz

**Data Availability Statement:** All relevant data are within the manuscript and its Supporting Information files.

## Abstract

The aim of this study was to determine the effect of autologous serum (AS) eye drops on the density of human leucocyte antigen (HLA)-DR-positive epithelial cells and Langerhans cells on the ocular surface of patients with bilateral severe dry eye disease (DED) due to graft-versus-host disease (GvHD) or Sjögren's syndrome (SS). The study was conducted on 24 patients (48 eyes). AS was applied 6–10 times daily for 3 months together with regular artificial tear therapy. HLA-DR-positive cells were detected by direct immunocytochemistry on upper bulbar conjunctiva imprints obtained before and after treatment. The application of AS drops led to a statistically significant increase in the mean density of aberrant HLA-DR-positive conjunctival epithelial cells (p < 0.05) and HLA-DR-positive Langerhans cells (p < 0.05) in the GvHD group. Aberrant HLA-DR-positive epithelial cells in the SS group were decreased non-significantly. All patients reported a significant decrease in the Ocular Surface Disease Index (p < 0.01), which indicates improvement of the patient's subjective feelings after therapy. There was an expected but non-significant decrease of aberrant HLA-DR-positive conjunctival epithelial cells in the SS group only. However, the increased density of HLA-DR-positive cells, indicating slight subclinical inflammation, does not outweigh the positive effect of AS in patients with DED from GvHD.

## Introduction

Autologous serum (AS) or allogeneic serum eye drops can be profitably used in the treatment of dry eye of various etiologies, including graft-versus-host disease (GvHD) and Sjögren's syndrome (SS), both frequent causes of dry eye disease (DED). [1–5]

The effectiveness of AS is mainly attributed to the presence of various substances, including growth and neurotrophic factors, bacteriostatic factors, fibronectin, and vitamins, almost all of

**Funding:** This work was supported by the research project IGA NT/12376-4, project BBMRI_CZ LM2015089 (KJ) and by Progres-Q25 (KJ,SK,VV).

**Competing interests:** The authors have declared that no competing interests exist.

which are also present at lower levels in normal tears. [6, 7] The main described clinical effects of AS are the improvement of tear film stability and ocular surface vital staining scores [8]; from the laboratory point of view, the main effects are decreased squamous metaplasia (i.e., increased epithelial and goblet cell density, decreased keratinization) and an antiapoptotic effect. [9–11]

Human leucocyte antigen (HLA)-DR, a membrane glycoprotein normally restricted to cells of the immune system, plays a major role in initiating the immune response to an antigen. [12] On the healthy ocular surface, HLA-DR is abundantly present in Langerhans dendritic cells, but its occurrence is rare in conjunctival epithelial cells. [13, 14] On the other hand, strong aberrant HLA-DR expression in conjunctival epithelial cells was observed in patients with DED [12, 15, 16], where it may reflect an inflammation and DED severity. [14, 17–20] In multi-center clinical trials, HLA-DR has served as a biomarker of ocular surface inflammation. [21] HLA-DR expression, particularly in DED due to primary SS, has been found together with an increase in the proinflammatory marker intercellular adhesion molecule 1 (ICAM-1) and the inflammatory mediator interleukin-6 (IL-6). [17, 18, 22, 23] Similarly, HLA-DR expression followed by increased CD8-positive lymphocytes has been shown in patients with chronic GvHD, with or without any effect on the ocular system. [24]

Aberrant HLA-DR expression (i.e., its expression under pathological but not physiological conditions) was first described by Hanafusa et al. in 1983 in thyrocytes in Grave's disease. [25] Subsequently, HLA-DR aberrant expression has been described in various cell types in a number of chronic or autoimmune diseases, e.g., primary biliary cirrhosis or biliary atresia (bile duct epithelium), active rheumatic carditis (valvular fibroblasts), Crohn's disease (esophageal epithelium), and diabetes mellitus (pancreatic β-cells). [26–30] It has been suggested that such aberrant expression could play a crucial role in immune system activation, inflammation, and the development of autoimmune disease. [31] Finally, aberrant HLA-DR expression has been described in the intestinal and dermal epithelia and in the epidermal keratinocytes of patients with GvHD. [32–34] Recently, aberrant HLA-DR expression has been found in the syncytiotrophoblast of pre-eclampsia placentae. [35] Deregulated HLA-DR expression in the epithelial conjunctival cells of patients with DED, but not that of healthy individuals, is essentially the same finding as the aberrant expression described in the cell types listed above, thus such HLA-DR expression in epithelial conjunctival cells can rightfully be considered aberrant as well.

In this study, we were interested in determining whether AS use in the most common scheme (3 months' application of 20% AS together with artificial tears) would affect the presence of HLA-DR-positive Langerhans and epithelial cells on the conjunctival surface of patients with DED.

## Materials and methods

### Patients

Twenty-four adult individuals (6 men, 18 women) with a mean age of 49 (range, 25–71) years with bilateral severe DED were eligible for enrollment in the study. The study followed the tenets set forth in the Helsinki Declaration and was approved by the local ethics committee (Ethics Committee of the General University Hospital, Prague). Written informed consent was obtained from all patients. Severe DED was defined by the following inclusion criteria: Schirmer test I (ST I) < 5 mm (measuring reflex tear secretion over 5 minutes while allowing natural blinking), tear film breakup time (tBUT) < 5 seconds, and subjective symptoms in the Dry Eye Severity Level adopted by the Dry Eye Workshop Committee. [36] The patients used the Ocular Surface Disease Index (OSDI) to assess DED-related symptoms before and after AS

application. [37] The cause of dry eye was attributed to chronic GvHD in 12 patients (6 men) and to primary SS in 12 patients (only women).

Sixteen patients were under the treatment with preservative-free artificial tear eye drops (nine from GVHD and seven from SS group respectively), and eight patients with artificial tears containing oxychloro complex (Purite) or Polyquad preservatives (four patients per each group). Seven patients, all from the GvHD group, were on systemic immunosuppressive therapy and antiviral therapy. In the SS group, two patients were receiving systemic corticosteroids. Patients who did not have any change in their anti-inflammatory and artificial tears regimen for at least two months prior to the application of AS and during the follow-up period were included in the study.

## AS eye drop preparation and application

AS was prepared from 40 ml venous blood; after centrifugation (3000 × *g* for 15 minutes), the serum was diluted with an isotonic buffered saline solution to 20% as previously described. [6, 11] The AS eye drops were aliquoted into 10-ml dark sterile vials to protect them from ultraviolet light (SANO, Dr. Kulich Pharma, Hradec Kralove, Czech Republic) and frozen at -20˚C. Serology tests (AIDS, hepatitis B and C, syphilis) and a microbiology control test were performed (all with negative results). Storage instructions were issued to the patients based on our previous protocol: unopened vials were to be stored in the freezer at -20˚C, the vial currently in use was to be stored in the refrigerator at 4–8˚C for seven days maximum, and proper handling procedures should be followed to avoid the risk of contamination. The maximum storage duration was four months.

For three months, the patients administered the AS eye drops approximately 15 minutes after the application of artificial tears. The number of AS applications depended on the number of artificial tear applications: most of the patients applied AS 6–10 times daily. The patients were instructed not to change the frequency of the treatment during the entire duration of the study even if they experienced subjective improvement of their symptoms.

## Impression cytology and HLA-DR detection

Impression cytology was carried out twice, i.e., before and after AS treatment, on the upper bulbar conjunctiva of both eyes. Biopore membranes (Millicell-CM, PICM01250, Millipore) were used after the three legs present on the plastic holder had been removed. Immediately after obtaining the imprints, the membranes were frozen and stored at -80˚C until used.

Immunocytochemistry was performed directly on the Biopore membranes based on the method originally described by Donisi and colleagues (2003) and later modified by Jirsova and coworkers (2006). [38, 39] Briefly, membranes with the harvested cells were fixed and released from their plastic holders in one step by 1-minute acetone treatment, and while still wet, placed cell side up on round 12-mm coverslips. The membranes remained perfectly flat and adhered to the coverslips during all subsequent steps of the staining procedure. Direct immunocytochemistry was performed using a sandwich method in which the cells on the membrane were kept between parafilm (Parafilm M, Sigma-Aldrich, St Louis, MO, USA) and the coverslip, thus allowing the use of only a single drop (50 μl) of each incubation solution. The cells were permeabilized with 0.2% Triton X-100 (Sigma-Aldrich) in phosphate-buffered saline (PBS). After three washes, the specimens were incubated for 1 h in dark with a fluorescein isothiocyanate (FITC)-conjugated monoclonal antibody against HLA class II DR (1:10, Acris Antibodies GmbH, Germany). The specimens on membranes were mounted between glass slide and coverslip using VectaShield mounting medium with propidium iodide (Vector Laboratories, Burlingame, CA, USA) to counterstain the nuclei. One day later, immediately before microscopic

evaluation, membrane transparency was induced by adding one drop of PBS between the slide and the membrane on the coverslip. An Olympus BX51 light microscope (Olympus, Tokyo, Japan) was used for evaluation at × 100–400 magnification. Ten to forty non-overlapping photographs (from image fields covering an area of 0.14 mm$^2$) were taken with a CCD-1300 camera (VDS Vosskühler GmbH, Germany).

The density of HLA-DR-positive epithelial cells and Langerhans cells was assessed independently by two researchers (KJ, VV) blinded to the experimental conditions using a NIS Elements image analysis system (Laboratory Imaging, Prague, Czech Republic). The HLA-DR-positive cells were clearly discernible from negative cells. Cells with typical dendriform or epithelial morphology were assessed separately. Due to the presumed increased density of epithelial cells as a consequence of decreased squamous metaplasia after AS application, [9–11] which would lead to a misleading increase in the number of positive cells, the percentage of HLA-DR-positive epithelial cells was calculated in 1200 cells from areas representing the diversity of the sample equally. Specimens with <50% confluence were excluded from the assessment.

## Statistical analysis

The data were log-transformed prior to statistical analysis. The estimated sample means of all studied variables (cells/mm$^2$) are reported as the geometric mean and its 95% confidence interval. To address the repeated measurements of the cell densities of each patient (right and left eye of every patient), the effect of each categorical factor (phase of study, diagnosis) and their interaction were assessed using repeated-measures analysis of variance. A generalized estimation equations model was used to identify statistically significant differences in the cell density values between all possible combinations of categorical factors (before vs. after AS application, diagnosis of GvHD vs. primary SS). Graphical analysis and the Shapiro-Wilk and Levene tests were used to examine the validity of the models. Statistical analyses were conducted using R language for statistical computing and graphics (R Core Team, 2013). P-values < 0.05 were considered statistically significant. To verify whether either age or gender may have an effect on the collected data, these two confounders were analyzed. To evaluate the effect of the age factor on GvHD vs. primary SS comparison, the mean, median and skewness of patient age of both groups were calculated in order to verify the normality of the sets and t-test used to evaluate the statistical difference. The gender factor was determined only for GvHD group (no males in SS group) by expressing the normalized percentual change of HLA-DR-positive epithelial density using formula 100*(A-B)/B, where A and B are HLA-DR-positive cell per mm$^2$ after and before AS application respectively. Student t-test was than applied to determine the difference.

## Results

In the overall patient population, the mean OSDI score (arithmetic mean and its 95% confidence interval) before AS application was 68.9 (64.2; 73.7), which decreased significantly to 57.6 (52.5; 62.8) after application (p < 0.001); in the GvHD group, the mean OSDI score of 71.0 (65.3; 76.7) decreased to 59.4 (53.1; 65.8) (p = 0.007), whereas that for the SS group decreased from 67.2 (59.7; 74.8) to 56.1 (48.0; 64.2) (p = 0.041).

Of 48 specimens, one was discarded from assessment due to the extent of confluence <50%. The confluence of most imprints was 70–90%. In total, 47 dry eye specimens were analyzed before and after AS application: 23 from the GvHD group and 24 from the primary SS group.

The following cell density values are reported as the absolute frequencies of positive cells/mm$^2$. The density (mean, 95% confidence interval) of HLA-DR-positive epithelial cells in all patients was 54.7 (30.4; 98.0)/mm$^2$ and 78.9 (44.2; 140.2)/mm$^2$ before and after AS application respectively. The overall difference for all patients was not statistically significant (p = 0.244);

the interaction between the phase of the study (before vs. after AS application) and the diagnosis group was statistically near significant (p = 0.056). When we assessed the HLA-DR-positivity as a number of positive cells per 100 cells, 27% and 31% HLA-DR-positive epithelial cells on average were detected before and after treatment, respectively. The mean densities of HLA-DR-positive Langerhans cells before and after AS application were 2.8 (1.8; 4.2)/mm$^2$ and 4.3 (2.7; 6.7)/mm$^2$, respectively, which was a statistically significant increase (p = 0.034) (Fig 1).

As the age might be an influencing factor when comparing the GvHD vs SS group, we tested the age of both groups for similarity. The mean, median and skewness for GvHD and SS were 49.1, 49 and -0.44, and 48.2, 49.5 and 0.43 respectively. Student t-test value was 0.94 showing the equality of age representation in both groups and validity of GvHD versus SS comparison.

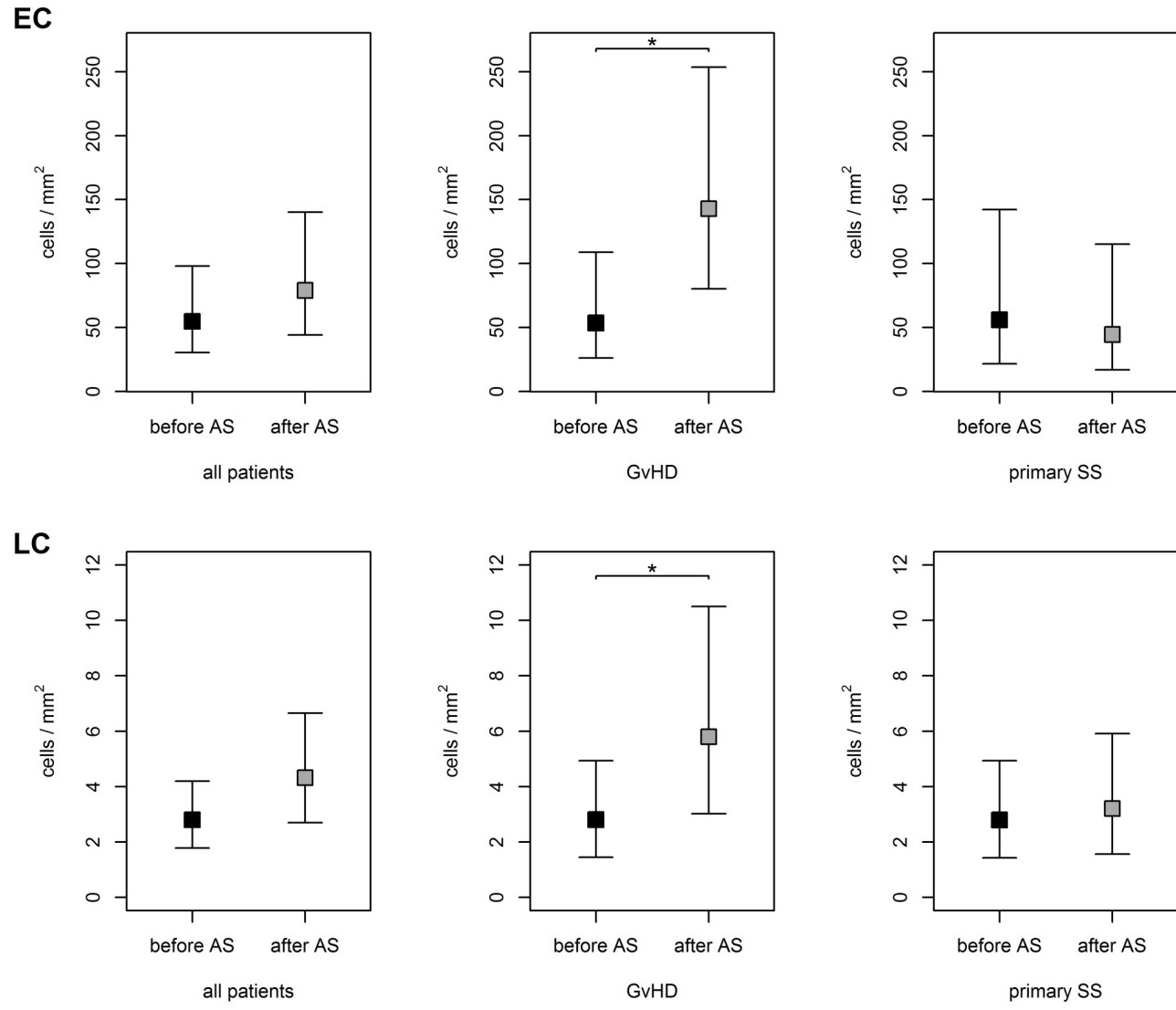

**Fig 1. The densities of HLA-DR-positive epithelial (EC) and Langerhans cells (LC) in dry eye patients.** The significant increase of both HLA-DR-positive epithelial and Langerhans cells densities was present in GvHD, but not in SS patients after the application of AS. * P value <0.05.

When we assessed the patients with GvHD separately, the mean density of HLA-DR-positive epithelial cells increased significantly (p = 0.049) post-treatment from 53.5 (26.0; 108.9)/mm$^2$ to 142.9 (80.3; 253.6)/mm$^2$ (percentages: from 27% to 40%). The density increased in 16 of 24 eyes. The pre- and post-treatment densities of HLA-DR-positive Langerhans cells were 2.8 (1.5; 4.9)/mm$^2$ and 5.8 (3.0; 10.5)/mm$^2$ (p = 0.042), respectively (Figs 1 and 2 and Table 1). The percentage increase in HLA-DR-positive epithelial cells did not differ significantly between men and women from GvHD group (p = 0.829).

In primary SS, the pre- and post-treatment densities of HLA-DR-positive epithelial cells were 55.9 (21.6; 142.2)/mm$^2$ and 44.5 (16.8; 115.1)/mm$^2$ (percentages: 26% and 23%), respectively. The density decreased in 13 of 24 eyes. The pre- and post-treatment densities of HLA-DR-positive Langerhans cells were 2.8 (1.4; 4.9)/mm$^2$ and 3.2 (1.6; 5.9)/mm$^2$, respectively, however none of these differences were significant (p = 0.689 and p = 0.674) (Figs 1 and 2).

The HLA-DR-positivity in epithelial cells was present particularly in the cell membranes and the cytoplasm (Fig 3a). The epithelial and Langerhans cells were in most cases clearly discernible based on their morphology (Fig 3a and 3b) but sometimes the intense signal of the epithelial cells was superimposed upon the normal background of the Langerhans cells. A strong signal was often present in squamous epithelial cells with a lower nuclear-cytoplasm ratio (Fig 3c). No positivity was observed in the goblet cells scattered throughout most of the specimens, particularly after AS treatment (Fig 3d). The Langerhans cells from most patients had shorter spurs (Fig 3e) compared to Langerhans cells from healthy conjunctiva (Fig 3f). Inflammatory cells, present mostly in infiltrates, exhibited a prominent signal.

## Discussion

The use of AS became accepted as a standard treatment of DED of various etiologies, including systemic autoimmune diseases. [40, 41] The aim of the present study was to determine the effect

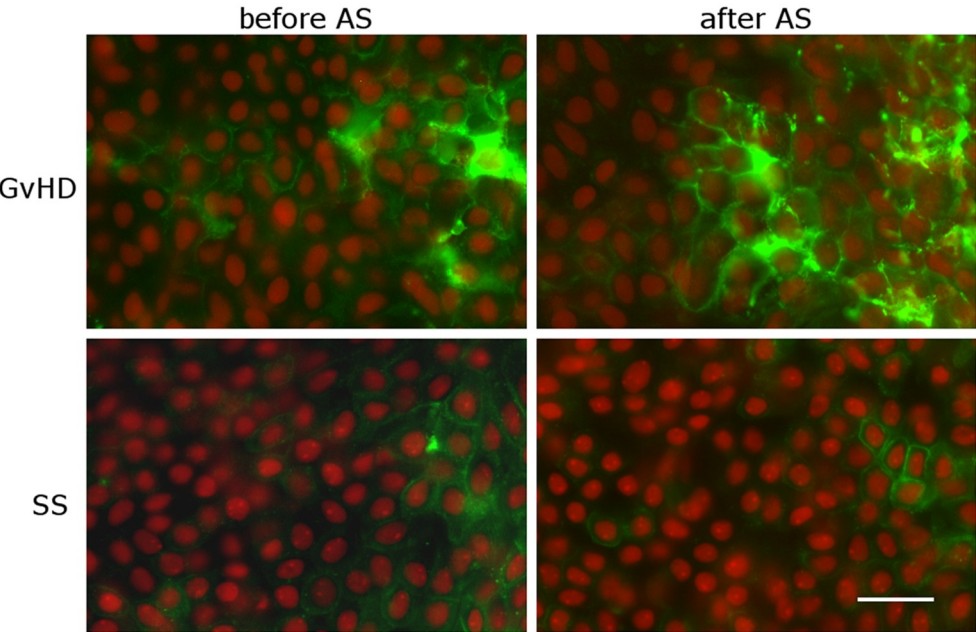

**Fig 2. Representative photographs of HLA-DR positivity on conjunctival imprints of patients suffering from GvHD or SS.** The number of aberrant HLA-DR-positive (green stain) conjunctival cells increase in patients suffering from GvHD, but not in SS patient's impression after the application of AS. Scale bar represents 50 μm.

**Table 1. The differences in epithelial and Langerhans cell densities between and within the phases of the study and the diagnosis groups (GvHD and SS patients).**

| Epithelial cells | | | | | |
|---|---|---|---|---|---|
| **Phase of study**[1] | **Diagnosis group**[1] | | | **Repeated measures ANOVA** | |
| | GVHD | Primary SS | p[2] | Factor | p[4] |
| Cells/mm$^2$ | | | p-value | | |
| Before AS | 53.5 (26.0; 108.9) | 55.9 (21.6; 142.2) | 0.857 | Phase of study | 0.244 |
| After AS | 142.9 (80.3; 253.6) | 44.5 (16.8; 115.1) | 0.059 | Diagnosis group | 0.394 |
| | | | | Interaction | 0.056 |
| p[3] | 0.049 | 0.689 | | | |
| **Langerhans cells** | | | | | |
| **Phase of study**[1] | **Diagnosis group**[1] | | | **Repeated measures ANOVA** | |
| | GVHD | Primary SS | p[2] | Factor | p[4] |
| Cells/mm$^2$ | | | p-value | | |
| Before AS | 2.8 (1.5; 4.9) | 2.8 (1.4; 4.9) | 0.984 | Phase of study | 0.034 |
| After AS | 5.8 (3.0; 10.5) | 3.2 (1.6; 5.9) | 0.294 | Diagnosis group | 0.530 |
| | | | | Interaction | 0.175 |
| p[3] | 0.042 | 0.674 | | | |

Data were log-transformed prior to Repeated measures ANOVA analysis.

[1] geometric mean and 95% confidence interval

[2] statistical significance of difference between GvHD and primary SS

[3] statistical significance of difference between particular phases of the study

[4] overall statistical significance of analyzed factors (phase of study and diagnosis groups and their interaction)

of AS treatment on the density of HLA-DR-positive Langerhans cells and aberrantly HLA-DR-positive epithelial cells on the conjunctival surface of patients with GvHD and SS DED. We expected the HLA-DR-expressing cells to decrease after therapy, which would correlate with a positive effect of AS. However, our findings do not follow our expectations completely. Although we found a significant decline of OSDI in both groups of patients and a non-significant decrease in aberrant HLA-DR expression in the epithelial conjunctival cells from the patients with SS, we also detected significant increase of HLA-DR-positive epithelial and Langerhans cells in the conjunctiva of the patients with GvHD.

The decrease of the aberrantly positive epithelial cells in the patients with SS to 82% of pre-treatment, has the same trend as in our previous study, in which HLA-DR-positive cells in patients with primary and secondary SS decreased to 58% of baseline following AS treatment. [11]

The significant elevation of aberrant HLA-DR expression in the conjunctival epithelium of the patients with GvHD, which was 204% and 148% of baseline (assessed as the density or percentage of positive cells), respectively, could be explained by the stimulation of subclinical inflammation induced by AS application in GvHD, but not in SS. The response of the GvHD Langerhans cells was similar to that of the GvHD conjunctival epithelial cells: the GvHD group had significantly higher HLA-DR-positive Langerhans cell density (up to 137% of baseline following treatment). In contrast, there was a smaller, non-significant change in the SS group (an increase of 28% of baseline).

The increase in aberrant HLA-DR expression in the GvHD conjunctival epithelial cells does not correlate with the improvement of their OSDI scores (p < 0.001), which was even greater than of the patients with primary SS (p < 0.05). The positive effect of AS on clinical and laboratory results of GvHD patients [2, 42] is consistent with the willingness of the majority of our patients (65%) to repeat the AS therapy. Similarly, improved symptoms, reflected by

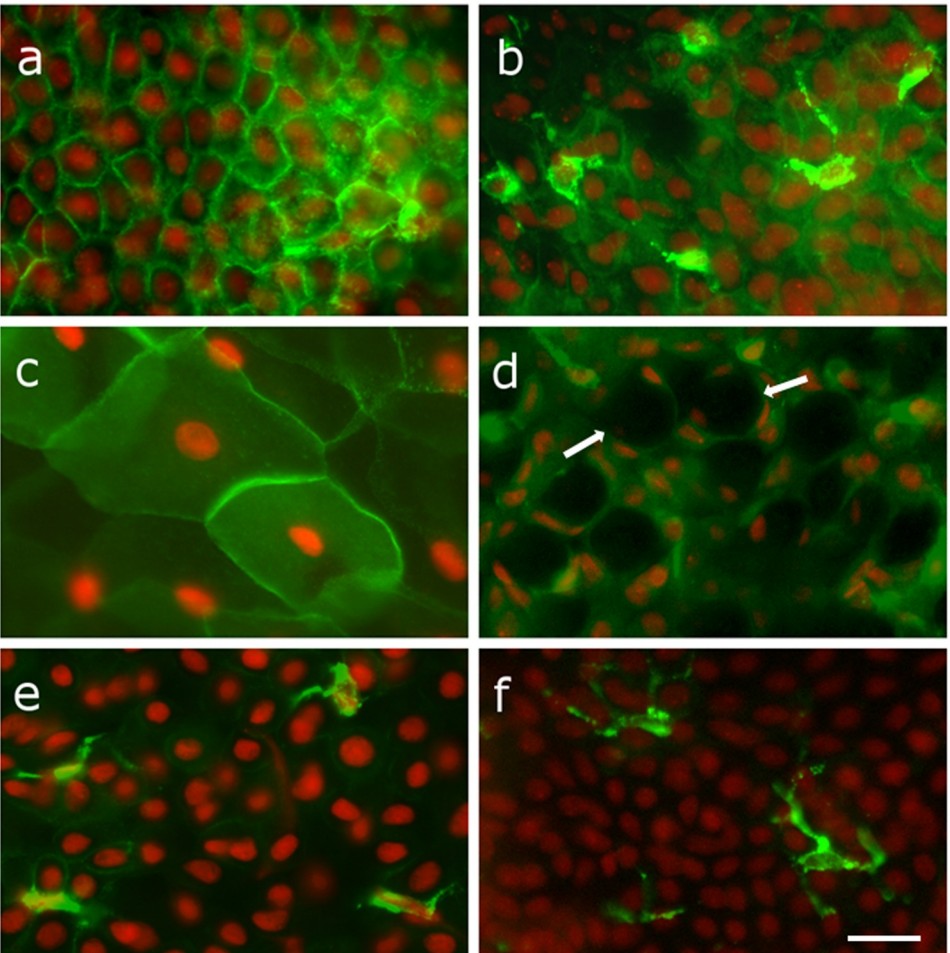

**Fig 3. Various types of positivity in epithelial and Langerhans cells in conjunctival imprints.** Cytoplasmic and cell membrane positivity in epithelial cells (a), positivity in Langerhans cells is clearly apparent based on their morphology from weakly stained epithelial cells (b), a strong signal in squamous epithelial cells (c), the lack of HLA-DR antigen expression in goblet cells (d), Langerhans cells exhibited short spurs in dry eye disease patients (e), Langerhans cells from healthy conjunctiva exhibited long spurs (f). Scale bar represents 50 μm.

the decrease of OSDI score, have been found after the application of both autologous and allogenic serum. [41, 43] These findings clearly demonstrate that the positive effect of AS unambiguously outweighs the negative impact of possible subclinical inflammation.

Recently, analysis of cells obtained by conjunctival impression cytology revealed that HLA-DR, IL-6, CXCR4, and CCL2/CCR2 mRNA levels are significantly increased in DED patients of various etiologies compared to controls. [44] Interferon (IFN)-γ, IL-1, IL-6, and tumor necrosis factor (TNF)-α, as well as chemokines (IL-8), are elevated in the tears of patients with SS compared to healthy subjects. [45, 46] The upregulation of several proinflammatory chemokines has also been detected in the conjunctiva of patients with GvHD. [47] HLA-DR expression in conjunctival epithelial cells mediates T lymphocyte homing, which in turn induces the apoptosis of cells bearing the HLA-DR antigen, as it has been shown for both GvHD and SS. [22, 23, 48, 49]

We suggest that the increased HLA-DR expression in the present study is tightly associated with the pathogenesis of GvHD, where aberrant HLA-DR expression has also been observed

in the intestinal and dermal epithelia as well as in epidermal keratinocytes. [32–34] Such aberrant expression has not been observed outside the conjunctiva in patients with SS. The application of AS on the ocular surface of patients with GvHD may stimulate the immune system and consequently promote the herein-described aberrant expression of HLA-DR.

Increased proinflammatory cytokines, particularly TNF-α and IL-6, have been found in the serum of patients with GvHD and SS. [50, 51] To the best of our knowledge, no comparative study measuring the levels of a set of proinflammatory factors in sera from GvHD and SS patients exists. Our suspicion that an elevated level of proinflammatory compounds stimulates HLA-DR expression in GvHD is supported by the significantly higher concentration of IL-17 found in GvHD serum compared to SS serum. [52] It is a question of whether the host cells in GvHD support this proinflammatory response after AS application or whether the recipient conjunctival epithelium responds with a stronger reaction following stimulation by the proinflammatory factors present in AS. The intensity of immunosuppressive or other systemic treatment of our patients did not correlate with the observed changes in the density of the HLA-DR-positive cells.

In summary, the improvement of the patient´s subjective feelings after AS therapy is reflected by decreased values of Ocular Surface Disease Index. There was also an expected but insignificant decrease of aberrant HLA-DR-positive conjunctival epithelial cells in patients with Sjögren's syndrome. The increased density of HLA-DR-positive cells in GvHD patients, which is a potential indicator of subclinical inflammation, should not be in our opinion a reason to discriminate these patients from AS application. Rather, it is prompting to compare various blood products for their efficacy.

## Supporting information

**S1 Data.**
(PDF)

**S2 Data.**
(PDF)

## Author Contributions

**Conceptualization:** Katerina Jirsova.

**Data curation:** Petra Seidler Stangova, Michalis Palos, Ivana Rybickova.

**Funding acquisition:** Katerina Jirsova.

**Investigation:** Katerina Jirsova, Petra Seidler Stangova, Gabriela Mahelkova, Sarka Kalasova, Ivana Rybickova.

**Methodology:** Katerina Jirsova, Ivana Rybickova.

**Project administration:** Viera Vesela.

**Supervision:** Katerina Jirsova.

**Visualization:** Viera Vesela.

**Writing – original draft:** Katerina Jirsova.

**Writing – review & editing:** Katerina Jirsova, Petra Seidler Stangova, Tor Paaske Utheim.

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
