## [Decision Letter · Decision Letter 0]

24 Dec 2019

PONE-D-19-27094

Aberrant HLA-DR expression in the conjunctival epithelium after autologous serum treatment in patients with graft-versus-host disease or Sjögren’s syndrome

PLOS ONE

Dear Assoc. Prof., Jirsova,

Thank you for submitting your manuscript to PLOS ONE. After careful consideration, we feel that it has merit but does not fully meet PLOS ONE’s publication criteria as it currently stands. Therefore, we invite you to submit a revised version of the manuscript that addresses the points raised during the review process.

The authors performed an observational study of subjects with dry eye either secondary to oGVHD or primary Sjogren's syndrome to correlate clinical symptoms with impression cytology of the conjunctiva and to determine the number of HLADR+ cells before and after treatment with autologous serum eye drops. The reviewers found some problems with the paper that the authors need to address.

1. It is noted that there are many confounders in study - sex, previous treatment, reliability of counting + HLADR cells from IC stained filters. These need to be evaluated.

2. It is not clear if the readers of IC were " blinded" to pre vs. post AS treatment when evaluating samples.

We would appreciate receiving your revised manuscript by Feb 07 2020 11:59PM. To enhance the reproducibility of your results, we recommend that if applicable you deposit your laboratory protocols in protocols.io, where a protocol can be assigned its own identifier (DOI) such that it can be cited independently in the future. For instructions see: http://journals.plos.org/plosone/s/submission-guidelines#loc-laboratory-protocols

We look forward to receiving your revised manuscript.

Kind regards,

Alexander V. Ljubimov, Ph.D.

Academic Editor

PLOS ONE

Journal Requirements:

2. Please provide additional details regarding participant consent. In the Methods section, please ensure that you have specified what type of consent you obtained (for instance, written or verbal) and whether the ethics committee approved this consent procedure. If verbal consent was obtained please state why it was not possible to obtain written consent and how verbal consent was recorded. If your study included minors, state whether you obtained consent from parents or guardians.

3. Please ensure that you have provided a limitations section in your manuscript. This section should discuss the absence of a control/placebo group in your study.

We noticed you have some minor occurrence of overlapping text with the following previous work, which needs to be addressed:

https://doi.org/10.3109/02713683.2013.824987

In your revision ensure you cite all your sources (including your own works), and quote or rephrase any duplicated text outside the methods section. Further consideration is dependent on these concerns being addressed.

Additional Editor Comments:

The authors performed an observational study of subjects with dry eye either secondary to oGVHD or primary Sjogren's syndrome to correlate clinical symptoms with impression cytology of the conjunctiva and to determine the number of HLADR+ cells before and after treatment with autologous serum eye drops. The reviewers found some problems with the paper that the authors need to address.

1. It is noted that there are many confounders in study - sex, previous treatment, reliability of counting + HLADR cells from IC stained filters. These need to be evaluated.

2. It is not clear if the readers of IC were " blinded" to pre vs. post AS treatment when evaluating samples.

Reviewers' comments:

Reviewer's Responses to Questions

**Comments to the Author**

1. Is the manuscript technically sound, and do the data support the conclusions?

Reviewer #1: Partly

2. Has the statistical analysis been performed appropriately and rigorously? 

Reviewer #1: Yes

3. Have the authors made all data underlying the findings in their manuscript fully available?

Reviewer #1: Yes

4. Is the manuscript presented in an intelligible fashion and written in standard English?

Reviewer #1: Yes

5. Review Comments to the Author

Reviewer #1: Obervational study of subjects with dry eyes either secondary to oGVHD or primary Sjogren's syndrome to correlated clinical symptoms with impression cytology of the conjunctiva to determine number of HLADR + cells before and after treatment with autologous serum eye drops. Many confounders in study- sex , previous treatment, reliability of counting + HLAdr cells from IC stained filters. Also not clear if readers of IC were " blinded" to pre vs post AS treatment when evaluating samples. Study is interesting, but given potential confounders may be difficult to reach a conclusion. Most interesting part of study is the use of aberrant HLADR staining for positivity of conj. epithelial cells- this topic could be explored further

6. PLOS authors have the option to publish the peer review history of their article (what does this mean?). If published, this will include your full peer review and any attached files.

Reviewer #1: No

---

## [Author Response · Author response to Decision Letter 0]

28 Feb 2020

February 4, 2020

Alexander V. Ljubimov, Ph.D.

Academic Editor

PLOS ONE 

Dear Dr. Ljubimov, 

We thank the reviewers for their helpful comments, and we have addressed the comments from the referees as follows:

PONE-D-19-27094

Response to Reviewers 

1

The authors performed an observational study of subjects with dry eye either secondary to o GVHD or primary Sjogren's syndrome to correlate clinical symptoms with impression cytology of the conjunctiva and to determine the number of HLADR+ cells before and after treatment with autologous serum eye drops. The reviewers found some problems with the paper that the authors need to address.

It is noted that there are many confounders in study - sex, previous treatment, reliability of counting + HLADR cells from IC stained filters. These need to be evaluated.

Potential confounder factors has been analyzed where possible. No statistical difference was found in the age of patients with GVHD and SS respectively (p=0,94). See Statistical analysis and Results sections in the manuscript. 

Due to the fact that group of SS involves just females, and GVHD patients included both males and females, statistical difference of HLA-Dr positivity was calculated separately for GVHD males and GVHD females. No significant difference was found. This information was included in the manuscript. 

The paragraph regarding treatment of patients (Material and Methods section) was rewritten separately for GVHD and SS patients respectively ((“Sixteen patients were under the treatment with preservative-free artificial tear eye drops (nine from GVHD and seven from SS group respectively), and eight patients with artificial tears containing oxychloro complex (Purite) or Polyquad preservatives (four patients per each group). Seven patients, all from the GvHD group, were on systemic immunosuppressive therapy and antiviral therapy. In the SS group, two patients were receiving systemic corticosteroids.”)). The therapy of dry eye disease is highly individualized, the broad spectrum of medical treatment does not allow statistical analysis. It should be, however, noted that therapy regimen was not changed for any patient involved in the study for at least two months prior to the application of AS and during the follow-up period, which we consider to be an important factor.

The sentence ”The HLA-DR-positive cells were clearly discernible from negative cells.” was added to Material and Methods section.

2. It is not clear if the readers of IC were " blinded" to pre vs. post AS treatment when evaluating samples.

Images were evaluated by two independent investigators blinded to the experimental conditions (patient identity, before and after AS specimens). Following sentence was added to the text.

“The density of HLA-DR-positive epithelial cells and Langerhans cells was assessed independently by two researchers (KJ, VV) blinded to the experimental conditions using a NIS Elements image analysis system (Laboratory Imaging, Prague, Czech Republic).” 

Yours sincerely, Katerina Jirsova

Assoc. Prof. Katerina Jirsova, Ph.D.

Laboratory of the Biology and Pathology of the Eye

Institute of Biology and Medical Genetics, First Faculty of Medicine, Charles University and General University Hospital in Prague, 

Albertov 4, 128 00 Prague 2, 

Czech Republic

Tel: 00420 224 968 006, E-mail: katerina.jirsova@lf1.cuni.cz

---

## [Editor Report · Decision Letter 1]

25 Mar 2020

Aberrant HLA-DR expression in the conjunctival epithelium after autologous serum treatment in patients with graft-versus-host disease or Sjögren’s syndrome

PONE-D-19-27094R1

Dear Dr. Jirsova,

We are pleased to inform you that your manuscript has been judged scientifically suitable for publication and will be formally accepted for publication once it complies with all outstanding technical requirements.

With kind regards,

Alexander V. Ljubimov, Ph.D.

Academic Editor

PLOS ONE

Additional Editor Comments (optional):

The authors adequately replied to the critique.
---

## [Editor Report · Acceptance letter]

9 Apr 2020

PONE-D-19-27094R1 

Aberrant HLA-DR expression in the conjunctival epithelium after autologous serum treatment in patients with graft-versus-host disease or Sjögren’s syndrome 

Dear Dr. Jirsova:

I am pleased to inform you that your manuscript has been deemed suitable for publication in PLOS ONE. Congratulations! Your manuscript is now with our production department. 

With kind regards,

on behalf of

Dr. Alexander V. Ljubimov 

Academic Editor

PLOS ONE